# A Differential Proteomic Approach to Characterize the Cell Wall Adaptive Response to CO_2_ Overpressure during Sparkling Wine-Making Process

**DOI:** 10.3390/microorganisms8081188

**Published:** 2020-08-04

**Authors:** Juan Antonio Porras-Agüera, Juan Carlos Mauricio, Jaime Moreno-García, Juan Moreno, Teresa García-Martínez

**Affiliations:** 1Department of Microbiology, Agrifood Campus of International Excellence ceiA3, C6 building, Campus de Rabanales, University of Córdoba, E-14014 Córdoba, Spain; b02poagj@uco.es (J.A.P.-A.); b62mogaj@uco.es (J.M.-G.); mi2gamam@uco.es (T.G.-M.); 2Department of Agricultural Chemistry, Agrifood Campus of International Excellence ceiA3, C3 building, Campus de Rabanales, University of Córdoba, E-14014 Córdoba, Spain; qe1movij@uco.es

**Keywords:** sparkling wine, yeast, cell wall, flocculation, protein, CO_2_ overpressure

## Abstract

In this study, a first proteomic approach was carried out to characterize the adaptive response of cell wall-related proteins to endogenous CO_2_ overpressure, which is typical of second fermentation conditions, in two wine *Saccharomyces cerevisiae* strains (P29, a conventional second fermentation strain, and G1, a flor yeast strain implicated in sherry wine making). The results showed a high number of cell wall proteins in flor yeast G1 under pressure, highlighting content at the first month of aging. The cell wall proteomic response to pressure in flor yeast G1 was characterized by an increase in both the number and content of cell wall proteins involved in glucan remodeling and mannoproteins. On the other hand, cell wall proteins responsible for glucan assembly, cell adhesion, and lipid metabolism stood out in P29. Over-represented proteins under pressure were involved in cell wall integrity (Ecm33p and Pst1p), protein folding (Ssa1p and Ssa2p), and glucan remodeling (Exg2p and Scw4p). Flocculation-related proteins were not identified under pressure conditions. The use of flor yeasts for sparkling wine elaboration and improvement is proposed. Further research based on the genetic engineering of wine yeast using those genes from protein biomarkers under pressure alongside the second fermentation in bottle is required to achieve improvements.

## 1. Introduction

The production of sparkling wines following the traditional method (or *Méthode Champenoise*) implies a characteristic stage where yeast cells are subjected to a second fermentation in sealed bottle and an aging period in contact with lees. This whole stage is known as setting the foam or second fermentation in bottle, and yeast cells must be able to cope with stress mainly caused by ethanol toxicity (10–12% *v*/*v*), low temperature (9–12 °C), nutrient starvation, and CO_2_ overpressure (6–7 bar). Moreover, the aging period is known to contribute considerably to the wine quality and organoleptic properties through the release of cell wall and intracellular compounds during autolysis [1,2,3,4].

The yeast cell wall is a dynamic macromolecular complex in which components (β-1,3 and β-1,6-glucans, mannoproteins, and chitin) are linked, forming a molecular network with several functions [5]. Numerous studies have reported the structural and morphological changes of the cell wall, mainly during aging [6,7,8,9]. These authors confirmed that although the cell wall structure showed folds and morphological changes, it remained unbroken in yeast cells at long-term aging. The cell wall is essential not only for maintaining the cell morphology during growth, mating, or sporulation, but also for dealing with stress conditions that destroy the cell integrity. Under stress conditions, cell wall composition varies in response to environment, and the existence of a cell response illustrates its dynamic nature [10,11]. This response to stress, which is known as the “compensatory mechanism”, has been characterized by an increase in the bulk of cell wall proteins, chitin content increase, glucans synthesis, and cell wall components’ redistribution and remodeling [12,13,14].

Among the cell wall components, mannoproteins and those involved in flocculation are the most relevant from the industrial and enological point of view. Mannoproteins represent between 35% and 40% of the cell wall, and numerous studies have associated these glycoproteins with the wine quality and organoleptic properties such as aroma [15], color [16], and foam [17]. In this context, the overproduction of mannoproteins has become one of the most desirable aspects for the yeast selection [16]. On the other hand, flocculation implies a nonsexual, homotypic, reversible, multivalent, and Ca_2_^+^-dependent process in which yeast cells aggregate, forming flocs. The flocculent capacity of yeast cells is considered a distinctive feature allowing a fast, cost-effective, and environmentally friendly way to remove yeast cells during wine clarification [18,19]. In addition, the higher ethanol stress resistance of *Saccharomyces cerevisiae* in biofilms provides an efficient ethanol fuel production, in respect to free cells, during industrial cell immobilization [20]. Among the genes that regulate flocculation, *FLO11* is the main flocculin in yeasts required also for cell adhesion, invasive and pseudohyphal growth, and biofilm formation [21,22,23]. This last process is carried out by a special type of yeasts known as flor yeasts, which are capable of forming a biofilm on the wine surface and assimilating ethanol under oxidative conditions [24].

In this study, a novel proteomic approach was developed to identify cell wall-related proteins with the aim of characterizing and comparing their response to CO_2_ overpressure along the second fermentation in two industrial *S. cerevisiae* strains (a sparkling wine strain P29 and a flor yeast G1 implicated in sherry wine production). Understanding of the yeast behavior and cell wall proteomic response to such special conditions would provide relevant insight into the yeast cell wall and would allow improving the industrial process of sparkling wine elaboration and second fermentation yeast strains.

## 2. Materials and Methods

### 2.1. Yeast Strains and Conditions

In this work, two industrial yeast strains were used: *Saccharomyces cerevisiae* P29 CECT 11770, a yeast strain commonly used in sparkling wine elaboration and isolated from INCAVI (Catalan Institute of Vines and Wines, Vilafranca del Penedès, Barcelona, Spain), and a flor yeast G1 ATCC MYA-2451, which is responsible for the biological aging of sherry wines and is isolated from a wine flor velum biofilm from Montilla-Moriles region, Spain. 

Previously, yeast strains were grown in Yeast Extract–Peptone–Dextrose medium (YPD, 1% yeast extract, 2% peptone, and 2% glucose) and later acclimated separately in a pasteurized must (Macabeo white grape variety with 174.9 g/L of sugar, 3.6 g/L total acidity, and pH 3.4) during 5 days at 22 °C. After reaching high cellular concentration, viability, and ethanol content (similar to base wine), “tirage” was carried out according to INCAVI in a commercial base wine (Macabeo and Chardonnay 6:4, 10.21% *v*/*v* of ethanol, 0.3 g/L of sugar, pH 3.29, 5.4 g/L of total acidity, and 0.21 g/L of volatile acidity) added with 22 g/L of sucrose and 1.5 × 106 cells/mL. Each yeast strain was fermented in two conditions: PC or pressure condition, using sealed bottles with a bidule and metal overcap; and NPC or non-pressure condition, using a perforated bidule. Sampling was performed at two points along the second fermentation: T1 or the middle of the second fermentation (3 bar), and T2 or one month after the end of the second fermentation (6.5 bar). Cells in sealed bottles were collected considering the pressure levels (T1: 3 bar and T2: 6.5 bar). Samples of control bottles without pressure were taken at the same times, just taking into consideration the similar values of ethanol content under both conditions. Culture mediums, study conditions, sampling, kinetics of second fermentation, and cell viability are described in detail by Porras-Agüera et al. (2019) [25].

### 2.2. Proteomic Analysis

Protein extraction and identification was developed using the methods described in Porras-Agüera et al. (2019) [25] and Ishihama et al. (2005) [26] for protein quantification. Proteins were properly separated using an OFFGEL High Resolution kit pH 3–10 (Agilent Technologies, Palo Alto, CA, USA) according to their isoelectric point. Once separated, these proteins were identified through mass spectrometry, after digestion with trypsin, using an LTQ Orbitrap XL (Thermo Fisher Scientific, San José, CA, USA) and a nano LC Ultimate 3000 system (Dionex Corporation, Sunnyvale, CA, USA). After identification, cell wall-related proteins were selected using the Gene Ontology section from the *Saccharomyces* genome database (SGD, http://www.yeastgenome.org/, access date: September 2019) and Uniprot (http://www.uniprot.org/, access date: September 2019) databases, which were later quantified through the protein content (mol %).

### 2.3. Confidence Parameters and Statistics

From the total of proteins, only those identified with a score >2 and observed peptides ≥ 2 were used in the analysis. These proteins were discussed according to their ratio content PC/NPC as over-represented (ratio ≥ 2) or under-represented (ratio ≤ 0.5). Moreover, proteins obtained with high contents and those found specifically in both strains were considered.

Cell wall proteins were sorted by biological processes (GO, Gene Ontology, Terms) using the tool “GO Term Finder” from the SGD database. For each GO Term, *p*-values and the FDR (False Discovery Rate) were calculated, and a *p*-value < 0.01 was considered at the time of selecting the GO Terms. The multiple sample comparison procedure (MSC) was performed using the software Statgraphics Centurion v. XVI (StatPoint Technologies, Warrenton, VA, USA), considering a confidence level of 95.0%, according to Fisher’s least significant difference (LSD) method. Furthermore, the software STRING v. 11.0 (https://string-db.org/) was used to build the protein interaction map. All data were normalized (square root) and auto scaled prior to analysis. 

## 3. Results and Discussion

In *S. cerevisiae* P29, a total of 594 proteins were detected under PCT1, 1517 were detected under NPCT1, 419 were detected under PCT2, and 392 were detected under NPCT2; whereas in *S. cerevisiae* G1, 568 proteins were obtained under PCT1, 1000 were obtained under NPCT1, 94 were obtained under PCT2, and 218 were obtained under NPCT2. In this study, 32 proteins specifically located in the yeast cell wall were identified in each condition and sampling time, as well as those proteins related to flocculation (Appendix A). In the case of cell wall-related proteins in the P29 strain, 12 were found under PCT1 (2.02%, 2.7 mol%), 6 were found under PCT2 (1.43%, 2.1 mol%), 26 were found under NPCT1 (1.71%, 2.7 mol%), and 7 were found under NPCT2 (1.79%, 3.2 mol%). On the contrary, higher frequencies and protein content were observed in flor yeast G1, especially at T2: 12 proteins were found under PCT1 (2.11%, 2.5 mol%), 13 were found under PCT2 (13.83%, 7.6 mol%), 22 were found under NPCT1 (2.2%, 2.2 mol%), and 16 were found under NPCT2 (7.34%, 9.5 mol%). The protein number in P29 decreased under both conditions along the second fermentation and the first month of aging, while the number, content, and frequency values in flor yeast at T2 considerably exceeded those in P29. In fact, in terms of content, the difference was 3.6-fold and 4.1-fold under PCT2 and NPCT2, respectively, in flor yeast G1. These results suggest a high requirement of cell wall proteins in flor yeast once second fermentation is over, which is probably for a later biofilm formation once nitrogen and fermentable carbon sources are limited [24,27]. However, the proteins associated with this process only appeared under NPCT1 in both P29 (5 proteins, 0.33%, 0.1 mol%) and G1 (3 proteins, 0.30%, 0.1 mol%), and under NPCT2 just in flor yeast G1 (1 protein, 0.46%, 0.3 mol%). 

In order to find the biological processes in which these proteins are involved, a GO analysis was carried out (Appendix A). In general, processes associated with cell wall organization or biogenesis and external encapsulating structure organization were highly enriched in both strains and conditions. Nevertheless, the negative regulation of cell aging was highlighted especially at T2 under both conditions and yeast strains (except under PCT2 in G1), although it was also was found under PCT1 in G1. Moreover, processes related to carbohydrate and polysaccharide metabolism were found in P29 and G1 under NPCT1, and also under this condition, a fungal-type cell wall (1- > 3)-β-d-glucan biosynthetic process was observed in flor yeast G1. As for the proteins associated with cell adhesion and flocculation, these were obtained only under NPCT1. Besides, processes such as invasive or pseudohyphal growth, which take place under glucose and nitrogen limitation, are highlighted in flor yeast G1.

To analyze the connections between the different proteins identified, a protein interaction network map, based on the 32 cell wall and flocculation-related proteins identified in total in *S. cerevisiae* P29 and G1, was built using the STRING v. 11.0 database (Figure 1). In the map, proteins are shown as nodes, and the edges represent the interactions between nodes. A PPI (protein-protein interaction) enrichment *p*-value < 1 × 10^−16^ indicates that the nodes are not random and the observed number of edges is significant. From the 32 nodes, a total of 135 edges were established. Nodes with different colors represent specific clusters obtained from an MCL (Markov Cluster Algorithm) clustering method. The strength of the connections is indicated by the edges thickness, the red nodes being those which showed the strongest interactions, and representing proteins required mainly for cell wall organization and structure. In addition, proteins required for cell separation and cytokinesis (clear green nodes) also obtained strong connections, along with the proteins involved in flocculation and response to glucose starvation (blue nodes). Green nodes represent proteins responsible for folding and response to stress, and just the protein Plb2p did not show interactions.

In general, most of the cell wall proteins were detected at T1 and especially under NPC in both yeast strains. However, although the protein number decreased in samples at T2 under both PC and NPC, and also PCT1, the content increased considerably under these conditions (Figure 2). For a better understanding, the most relevant processes in which cell wall proteins in both strains are involved are described below. Furthermore, the over and under-represented proteins under PC, as well as those found specifically in a yeast strain and proteins that obtained high content, have been discussed in detail.

### 3.1. Glucan Processing and Remodeling

The main core of the cell wall is formed by β-1,3-glucan chains connected to β-1,6-glucan polymers. This polysaccharides network is continuously subjected to remodeling by numerous enzymes, allowing yeast cells to grow, bud, and deal with lysis [28]. In this study, a high amount of enzymes involved in glucan processing were identified in both strains. Among them, there is Exg2p, an exo-1,3-β-glucanase with a glycosylphosphatidylinositol (GPI) anchor attachment site required for β-glucan assembly [29], and the protein Scw4p, which is highlighted in flor yeast G1 to be detected 2.4 and 3-fold under PC at T1 and T2, respectively (Table 1). Even though Scw4p has been associated with the glucanases, studies by Capellaro et al. (1998) [30] reported that it may have an involvement in mating. Furthermore, deletion of this gene has been related to abnormal morphology, affecting the β-1,3-glucan and mannoproteins network, as well as increasing chitin content [31]. As it is observed in Figure 2, the protein content of Scw4p showed a different pattern, decreasing and increasing in P29 and G1, respectively, along the second fermentation and the first month of aging. This behavior might be explained due to the increase in pressure levels (2-fold at T2) and typical stresses as high ethanol content or starvation, which could compromise the cell wall integrity. Apart from the detection of Scw4p, the protein Scw10p was found in both strains, showing decreases in content under PC, which is more remarkable in flor yeast G1 at T2 (Figure 2). While the expression of *SCW4* is constitutive, the gene *SCW10* is cell-cycle regulated [32]. Deletions of these homologous proteins (sharing 63% of amino acids) have resulted in cell wall changes and demonstrated their role in the mating process [30]. Besides, these proteins have been suggested to participate in concert with other cell wall proteins to maintain cell wall integrity [33]. The high requirement of enzymes in flor yeast responsible for cell wall remodeling, especially glucans, would allow yeast cells to make the cell wall rigid, tolerate the stress conditions, and avoid cell lysis [12,34].

Other relevant hydrolases identified in this study were the two major proteins of the cell wall, the endo-β-1,3-glucanase Bgl2p and the exo-β-1,3-glucanase Exg1p [35,36]. These proteins were found with high content in both strains, especially at T2 under both conditions. However, this content was different depending on the yeast strain, since while in P29, Bgl2p was more abundant under NPCT2, in G1, it stood out under PCT2; and the same behavior was observed for Exg1p. This can be better appreciated in Figure 2, where the content of both Bgl2p and Exg1p increased lightly under PCT2 (versus NPCT2) in flor yeast G1, and on the contrary, in P29, their content under PCT2 showed a considerable drop of 0.25 and 0.24 mol%, respectively. Deletions of both genes *BGL2* and *EXG1* have been reported to increase the chitin and glucan levels in the cell wall, respectively [37,38]. Moreover, Bgl2p has been implied in the incorporation of GPI-anchored cell wall proteins [39] and also in the limitation of the reproductive life span during aging [40]. Cell wall degradation, which takes place during the autolysis process, is carried out by numerous hydrolytic enzymes, of which the glucanases are the most relevant. Although autolysis has not been observed until 3–6 months of aging in sparkling wine with conventional strains [1], the observed increase of glucanases under pressure conditions in flor yeast G1 might promote an earlier release of cell wall components and therefore shorten the period of aging under lees. 

### 3.2. Glucan and Chitin Assembly

Once cell wall components are synthetized, these are assembled and cross-linked to the cell wall due to the action of different glycoproteins and enzymes. In this context, the glycoprotein Kre9p was identified exclusively in P29 under NPCT1. *KRE9* encodes an *O*-glycoprotein reported to be involved in β-1,6-glucan synthesis and assembly [41]. These authors confirmed that the disruption of this gene results in serious growth impairment and an altered cell wall containing less than 20% of β-1,6-glucan. In addition to this protein, those belonging to the *GAS* family (Gas1p, Gas3p, and Gas5p) were found in both strains (under PCT1 and NPCT1 in P29, and under all conditions in G1). According to Figure 2, the only difference was observed in Gas1p, whose content decreased at both T1 and T2 in flor yeast G1, while it increased just at T1 in P29. The results obtained by Matsushita et al. (2017) [42] support that the use of strains overexpressing the gene *GAS1* have several advantages for fermentation processes under stress conditions, especially for low-pH conditions. These proteins are known to be β-1,3-glucanosyltransferases required for the maintenance and formation of β-1,3-glucan [10,43,44]. Based on their expression patterns, they appear to play partially overlapping roles throughout the development: whereas *GAS1* and *GAS5* are induced during vegetative growth, *GAS2* is expressed exclusively during sporulation and is required for normal spore wall formation [45]. On the other hand, the protein Crh1p and the chitin transglycosylase Crh1p was found in both strains (under NPCT1 in P29, and under all conditions in G1; Appendix A). It functions in the transfer of chitin to β-1,6 and β-1,3 glucan in the cell wall [46], and it is known to be induced under cell wall stress [47]. From an industrial point of view, the addition of chitin has been demonstrated to reduce wine haze formation and improve the clarification process [48]. Moreover, the presence of chitin attached to glucans in the cell wall has been observed under stress conditions through the action of Crh1p. This component accumulates as much as 10 times more in cells with mutations affecting the synthesis of glucans, mannoproteins, or glycoproteins, in order to compensate for the cell integrity [49]. Therefore, the activation of glucan and chitin synthesis could be induced as a response to cell wall stress via the cell wall integrity pathway [13,50,51]. 

### 3.3. Mannoproteins

Apart from glucan and chitin, mainly mannoproteins represent between 35% and 40% of the cell wall, and they are covalently joined to the β-1,3-glucan network [10]. Among them, mannoproteins belonging to the *PIR* family (Pir1p, Hsp150p/Pir2p, Pir3p, and Cis3p/Pir4p) were detected in both strains; however, they were more abundant in terms of content in flor yeast G1 (Appendix A). The authors revealed that these yeast genes are homologous, containing internal tandem repeats of amino acids, and *PIR1* and *PIR2* were observed to participate during heat shock tolerance [52,53]. The mannoprotein Cis3p/Pir4p, exclusively located in the bud scars of vegetative cells [54,55], experimented a marked decrease in content at T2, comparing both PC and NPC (Figure 2). As for the rest, it highlighted both the increase of Hsp150/Pir2p and the decrease of Pir1p in flor yeast under PCT2 (Figure 2). Moreover, Pir3p was found with high content in flor yeast G1 under both PCT2 and NPCT2 (Appendix A), although it did not show great differences in content. These PIR proteins have been associated with numerous functions and processes such as cell wall stability and synthesis, and also heat and nutrient stress [56,57,58]. In addition to these mannoproteins, others such as Ccw14p, Cwp1p, Dan4p, and Pst1p were found in both strains. The covalently linked cell wall mannoprotein Ccw14p [59,60] was found in both strains, although its content was more relevant in flor yeast G1 (only identified at T1 under both conditions). Furthermore, deletion of this gene confirmed their role in biofilm formation in flor yeast G1, showing a decrease of the biofilm weight and cell adhesion [61]. Cwp1p localizes to the birth scars of daughter cells [62], and Dan4p was obtained in the two yeast strains and specifically in P29, both under NPCT1, respectively. Studies by Abramova et al. (2001) [63] confirmed that *CWP1* is down-regulated under anaerobic conditions, which agree with our results, since this protein was not found under PC (Appendix A). Lastly, the mannoprotein Pst1p, which is secreted by yeast-regenerating protoplasts [64], was found over-represented in P29 (Table 1), and it also stood out in flor yeast G1 in terms of content under PC (Figure 2). This protein has been observed to be important for cell wall integrity [65] and during the response to cell wall damage, along with Cwp1p [66].

The high amount of mannoproteins detected, some of them under pressure conditions, and more abundant in flor yeast G1 could be interesting from an industrial and enological point of view. These glycoproteins have been reported to positively affect the wine quality and organoleptic properties, improving parameters such as the wine aroma [15], color [16], and foam [17]. In this context, the overproduction of mannoproteins represents one of the most desirable aspects for the yeast strains selection. The abundance of mannoproteins observed in flor yeast may open a door to the use of this type of yeast in the sparkling wine elaboration or improvement. Besides, the accumulation of mannoproteins in the cell wall has been recently associated with enhanced stress resistance and fermentation performance [67].

### 3.4. Cell Separation

The enzymes required for cell separation and cytokinesis located in the cell wall (Figure 1) were detected also. The endoglucanase Egt2p stood out to be found specifically in flor yeast G1 under NPCT1. This protein has been associated with cell separation during the cell cycle after cytokinesis [68]. However, studies carried out by Pan and Heitman (2000) [69] revealed a new role of this protein in pseudohyphal growth, which could give us an insight into the yeast behavior, since in addition, this result is in accordance with the GO Terms “invasive filamentous growth” and “pseudohyphal growth” detected under NPCT1 in flor yeast (Appendix A). Another endoglucanase such as Dse4p, located in the cell wall and required for cell separation [70], was found under PCT1 and NPCT1 in P29, and under NPCT1 and NPCT2 in G1. The content of this protein decreased significantly in both strains (Appendix A), although this difference in content was higher in P29 (Figure 2). Furthermore, Sun4p, a member of the SUN family of proteins, is involved in the remodeling of the yeast cell wall and cell septation process during the various phases of yeast culture development and under various environmental conditions [71,72], and it was identified only under NPCT1 in both yeast strains (Figure 2). Additionally, the glucosidase Scw11p also was detected in both strains, and it seems to play a role in cell separation and conjugation during mating [30]. The detection of these proteins may indicate that yeast cells are dividing, and despite the fact that nutrients are available for yeast cells at T1, some cells of the colony could exhibit invasive and pseudohyphal growth. This would require a coordinated cell wall synthesis and remodeling in order to deal with the changes in cell morphology.

### 3.5. Proteins Related to Flocculation, Cell Adhesion, and Biofilm Formation

The selection of wine yeasts with flocculent capacity represents a desirable factor in sparkling wine elaboration, since this process would allow a fast clarification of fermenting product, thus reducing time and production costs [73]. The proteins required for this process, along with others involved in cell adhesion and biofilm formation, were found in both yeast strains and mainly under NPCT1 (Appendix A). Among them, Flo11p stands out as the main flocculin responsible for cell adhesion-related phenotypes in *S. cerevisiae* and whose analysis revealed complex mechanisms of genetic regulation [21,74]. However, this protein was not relevant in terms of content under PC (Figure 2), since it only was observed under NPCT1 in both strains and NPCT2 just in flor yeast G1. Protein kinase Snf1p and the subunit beta-3 Gal83p were identified in both strains. Studies by Kuchin et al. (2002) [75] showed evidence that Snf1p kinase regulates the transcription of *FLO11* during pseudohyphal growth and biofilm formation in response to glucose limitation. Moreover, it has been observed that the interaction of Snf1p–Gal83p (Figure 1) is required for invasive growth through *FLO11* activation [76]. Apart from these proteins, others such as the heat shock protein Hsp12p and the transcriptional regulator Snf2p were found just in *S. cerevisiae* P29 under NPCT1 (Appendix A). While the first one is responsible for membrane organization during stress [77] and is essential for biofilm formation [78], the second one is a catalytic subunit of the SWI/SNF chromatin-remodeling complex involved in *FLO11* activation [21]. Biofilm formation is known to be induced under nutrient limitation as glucose or nitrogen and oxidative conditions, which is in agreement with the detection of these proteins under NPC. The switch of fermentative to oxidative metabolism by flor yeasts is essential to allow cells to remain at the wine surface and metabolize ethanol into acetaldehyde [24]. The presence of proteins involved in cell adhesion and flocculation under NPCT1 in both strains could indicate that cells are forming flocs under these conditions. On the other hand, the detection of Flo11p in flor yeast G1 with high content under NPCT2 suggests that this yeast strain is developing a biofilm formation phenotype. 

### 3.6. Other Cell Wall Proteins

Apart from the cell wall proteins mentioned above, others participating in processes such as folding (Ssa1p and Ssa2p), lipid metabolism (Plb2p), and two proteins with unknown specific function (Ecm33p and Sim1p) were also detected in this study. The ATPase Ssa1p and the ATP-binding protein Ssa2p, both belonging to the HSP70 protein family [79], were relevant in both strains. These proteins were found to be over-represented in both yeast strains: Ssa1p just in P29 and the two in flor yeast G1 (Table 1). Since the main function of these proteins is to serve as molecular chaperones, binding newly translated proteins to assist in proper folding and prevent aggregation/misfolding [80], their presence under PC may be explained as a response to cell wall damage and stress. On the other hand, the lysophospholipase 2 or Plb2p, which is required for lipid metabolism [81], appeared exclusively in P29 just at T1 under both conditions (Appendix A). Ethanol stress has been proposed as the main factor to activate lipid membrane remodeling, in order to increase its stability and resistance [82]. Additionally, the cell surface glycosylphosphatidylinositol (GPI)-anchored protein Ecm33p, which is required for proper cell wall integrity and for the correct assembly of the mannoprotein outer layer [65,83,84], and the protein Sim1p—a member of the SUN family of proteins probably with a role in DNA replication [85]—were found: the first one over-represented in P29 (Table 1) and the second one in both strains, although the differences in content were not relevant (Figure 2).

## 4. Conclusions

According to the results, pressure seems to affect the number of cell wall proteins, mainly in *S. cerevisiae* P29. The results obtained in flor yeast G1 under CO_2_ overpressure conditions agree with those observed during a typical cell wall response to stress, and the abundance observed in mannoproteins makes this type of yeasts an interesting and innovative option for the improvement and elaboration of new sparkling wines. On the other hand, in *S. cerevisiae*, P29 stood out among the cell wall proteins responsible for glucan assembly and lipid metabolism. On the contrary, those proteins related to cell adhesion phenotypes (flocculation, cell adhesion, and biofilm formation) were not relevant under pressure, being observed exclusively in open bottles in both strains. Over-represented proteins under pressure were involved in cell wall integrity (Ecm33p and Pst1p) and folding (Ssa1p) in *S. cerevisiae* P29, and glucan remodeling (Exg2p and Scw4p) and folding (Ssa1p and Ssa2p) in flor yeast G1. The genes that codify these proteins may be interesting for exploration in the search for mechanisms involved in endogenous CO_2_ overpressure adaptation in sparkling wine yeasts, and the other hand, these target proteins might be involved in an accelerated aging process.

## Figures and Tables

**Figure 1 microorganisms-08-01188-f001:**
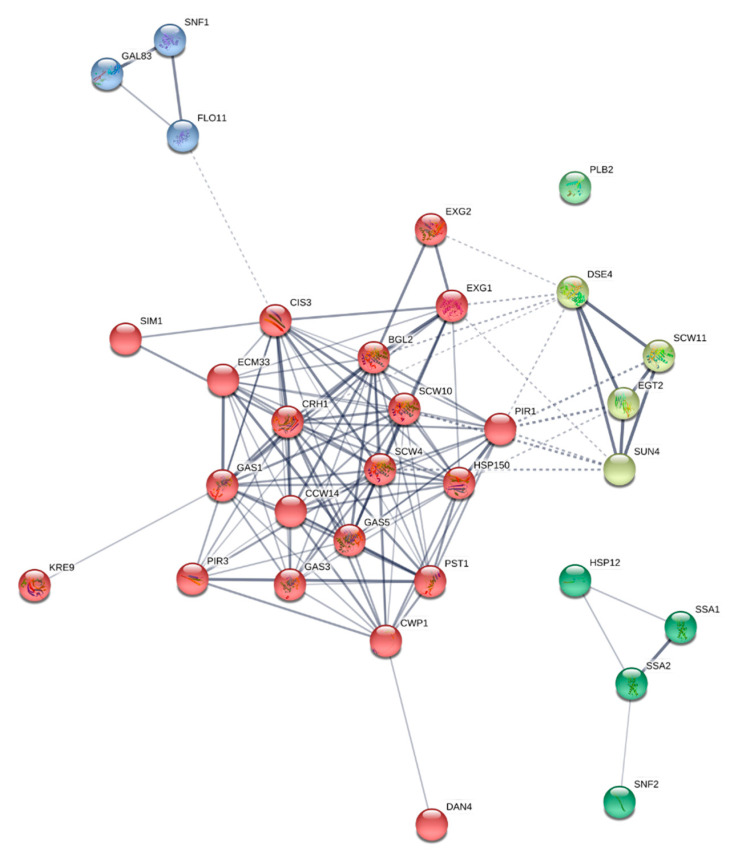
Protein interaction network map built using the STRING v. 11.0 database and based on the 32 cell wall-related proteins identified in total in *S. cerevisiae* P29 and G1. Proteins are shown as nodes, and the edges represent the interactions between nodes. The strength of the connections is indicated by the edges thickness. Nodes with different colors represent specific clusters obtained from an MCL (Markov Cluster Algorithm) clustering method (four clusters: blue, red, light green, and dark green). PPI (protein-protein interaction) enrichment *p*-value < 1 × 10^−16^.

**Figure 2 microorganisms-08-01188-f002:**
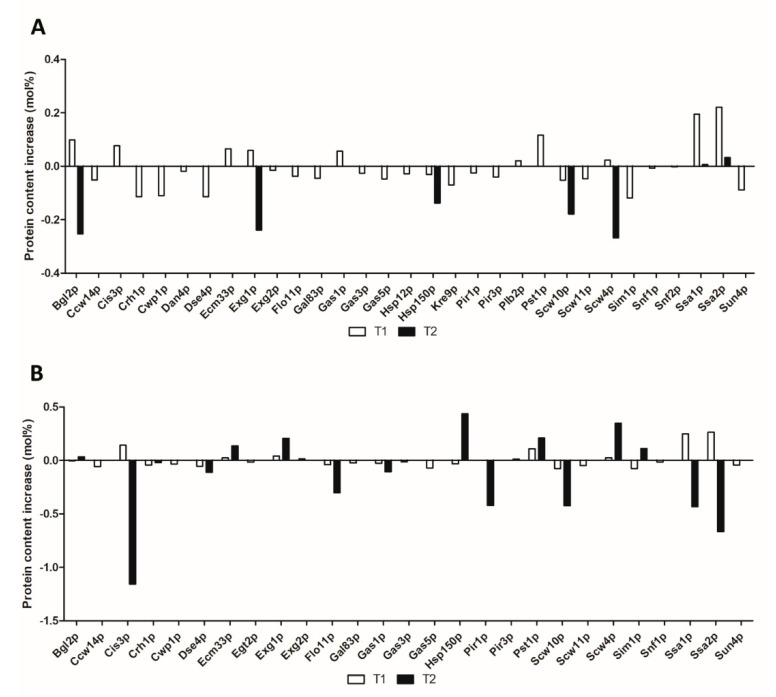
Protein content increases (mol%) observed in (**A**) *S. cerevisiae* P29 and (**B**) *S. cerevisiae* G1, under pressure conditions (PC) compared to non-pressure condition (NPC). T1 (middle of the second fermentation), T2 (one month after it).

**Table 1 microorganisms-08-01188-t001:** Over-represented proteins detected under pressure conditions (PC) in both yeast strains *S. cerevisiae* P29 and G1, at the middle of the second fermentation (T1) and one month after it (T2). Molecular function and fold change in brackets are shown. Only proteins with fold changes of protein content ≥ 1.8 are shown. GPI: glycosylphosphatidylinositol.

Yeast Strains	*S. cerevisiae* P29				*S. cerevisiae* G1			
Conditions	T1	Funcion	T2	Function	T1	Function	T2	Function
**Protein**	Ecm33p (2.1)	GPI-anchored protein	-	-	Cis3p (1.8)	Mannoprotein	Hsp150p (1.8)	O-mannosylated heat shock protein
	Gas1p (1.8)	β-1,3-glucanosyltransferase	-	-	Exg2p (2.4)	β-glucan assembly	Pst1p (1.8)	Cell wall protein with GPI (Glycosylphosphatidylinositol)-attachment site
	Pst1p (2.3)	Cell wall protein with GPI-attachment site	-	-	Pst1p (1.9)	Cell wall protein with GPI-attachment site	Scw4p (3)	Cell wall protein with similarity to glucanases
	Ssa1p (2)	Protein folding	-	-	Ssa1p (2.1)	Protein folding	-	-
	Ssa2p (1.9)	Protein folding	-	-	Ssa2p (2.1)	Protein folding	-	-

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
