# Peer review of "A Differential Proteomic Approach to Characterize the Cell Wall Adaptive Response to CO2 Overpressure during Sparkling Wine-Making Process"

_microorganisms, 2020, doi:10.3390/microorganisms8081188_

Round 1

Reviewer 1 Report

The results of the Manuscript “A Differential Proteomic Approach to Characterize 2 the Cell Wall Adaptive Response to CO2 Overpressure During Sparkling Wine Making Process” are related to a proteomic approach to study the response of cell wall related proteins to the CO2 pressure during the second fermentation step of sparkling wines.

The novelty is linked with the fact that authors used a proteomic approach to explain the behaviour of cell wall proteins using 2 different Saccharomyces cerevisiae strains. Authors concluded that the use of flor yeasts would be promoted for the improvement of the elaboration of sparkling wines.

Sentences and questions/remarks:

  • Line 89. Regarding T2, what does exactly mean “one month later”. Since the T1 time represented ½ of the second fermentation, T2 should be referred to the whole process of second fermentation.
  • Sentence in lines 142-143 should be re-written in order to clarify it.
  • In figure 1, maybe authors could choose different colours instead of two different types of green to make the figure clearer. In this figure the protein showing no interactions is Plb2p (in the text) and PLB2 in the figure. This fact should be solved.

Interesting conclusions were obtained for the enzymes conditioning glucan processing. The pressure predominantly impacts the number of cell wall proteins in P29 strain.

In general, the paper subjected to review has a consistent experimental design to achieve the objectives pursued with interesting results followed by well-founded discussions.

I consider that this paper should be accepted with minor revisions.

Author Response

The results of the Manuscript “A Differential Proteomic Approach to Characterize the Cell Wall Adaptive Response to CO2 Overpressure During Sparkling Wine Making Process” are related to a proteomic approach to study the response of cell wall related proteins to the CO2 pressure during the second fermentation step of sparkling wines.

The novelty is linked with the fact that authors used a proteomic approach to explain the behaviour of cell wall proteins using 2 different Saccharomyces cerevisiae strains. Authors concluded that the use of flor yeasts would be promoted for the improvement of the elaboration of sparkling wines.

-Sentences and questions/remarks:

  • Line 89. Regarding T2, what does exactly mean “one month later”. Since the T1 time represented ½ of the second fermentation, T2 should be referred to the whole process of second fermentation.

Samples were taken at two times along the second fermentation: at the middle of the process (T1, 3 bar), and one month after the completion of the second fermentation (T2, 6.5 bar).

The Materials and Methods has been modified in order to clarify this question.

  • Sentence in lines 142-143 should be re-written in order to clarify it.

The sentence was modified and clarified.

  • In figure 1, maybe authors could choose different colours instead of two different types of green to make the figure clearer. In this figure the protein showing no interactions is Plb2p (in the text) and PLB2 in the figure. This fact should be solved.

Yes, I can understand the possible confusion, but the software used for this Figure only admits the gene name in order to perform the interaction map.

Regarding the colour, once the interaction map is performed and the genes are grouped in clusters, the software does not allow you to modify the colour of the clusters. They are predetermined. The four colours have been described in the figure.

Interesting conclusions were obtained for the enzymes conditioning glucan processing. The pressure predominantly impacts the number of cell wall proteins in P29 strain.

In general, the paper subjected to review has a consistent experimental design to achieve the objectives pursued with interesting results followed by well-founded discussions.

I consider that this paper should be accepted with minor revisions.

We appreciated very much the reviewer´ comments.

Reviewer 2 Report

Why did you choose a flor yeast G1ATCCMYA-2451? Is there a specific reason?

Author Response

-Why did you choose a flor yeast G1ATCCMYA-2451? Is there a specific reason?

Flor yeast G1 was isolated from a wine flor velum biofilm from Montilla-Moriles region 30 years ago. This yeast strain is a typical flor yeast and has been well characterized by our research group. Firstly, we wanted to know the behaviour of cell wall proteome under second fermentation conditions, since this wine yeast is commonly used in other wine elaboration process with implies other enological conditions. Secondly, due to its interesting properties for enology such as high ethanol tolerance and flocculation capacity, we are also testing its possible use for sparkling wine improvement and production.

Thank you for the comment.

Reviewer 3 Report

The manuscript by Porras-Aguera et al. entitled “A differential proteomic approach to characterize the cell wall adaptive response to CO2 overpressure during sparkling wine making process” examines the levels of cell wall proteins in two yeast strains used for wine making under different conditions. While I can appreciate that this topic may be of interest to some researchers, I do not believe that the manuscript is comprehensive enough to constitute a standalone paper.

Major points

  1. This manuscript must undergo professional English editing prior to publication
  2. Section 2.2: more detail must be added to this methods subsection, it is unclear what type of experiment the authors did. In looking up the cited papers it appears that they did mass spectrometry – however this needs to be described in the current manuscript as it was not even mentioned.
  3. It is insufficient to perform mass spectrometry without examining select protein levels by western blot. The authors need to choose a subset of proteins to validate the expression of by western blot analysis. It should not be difficult to obtain cerevisiae antibodies to some of these proteins.
  4. From the way the manuscript is written, it appears as though the actual experiments were published in (Porras-Aguera et al., 2019 Microorganisms; https://doi.org/10.3390/microorganisms7110542) and that this paper is simply an additional analysis of this dataset with absolutely no new experiments. This does not constitute enough for a separate manuscript and the authors must greatly expand this paper in order for it to be suitable as a standalone manuscript. Perhaps the authors could choose some key genes of interest, knock them out and examine the effects on wine production. There are countless ways this manuscript can be expanded and I will not list them all, however this manuscript must be expanded somehow. The authors can use the mass spec results to inform their further analyses.

Minor points

  1. CO2 in title should either be formatted with a subscript “2” or spelled out as carbon dioxide
  2. Line 27: “Flocculation-related protein was not relevant under pressure conditions”, what is meant by not relevant?
  3. Line 67: “a first proteomic approach” should be “a novel proteomic approach”
  4. Line 88: “o” should be “or”
  5. Lines 96-100: paragraphs must be at least 3 sentences in length
  6. Line 104: “down-represented” should be “under-represented”
  7. Line 106: the “GO” acronym must be defined (“Gene ontology”)
  8. Reference to the “p” in “p-values” and “p < 0.XXX” must be italicized

Author Response

We appreciated very much the reviewer´ comments.

The manuscript by Porras-Aguera et al. entitled “A differential proteomic approach to characterize the cell wall adaptive response to CO2 overpressure during sparkling wine making process” examines the levels of cell wall proteins in two yeast strains used for wine making under different conditions. While I can appreciate that this topic may be of interest to some researchers, I do not believe that the manuscript is comprehensive enough to constitute a standalone paper.

Major points

-This manuscript must undergo professional English editing prior to publication

The English of the manuscript was checked.

-Section 2.2: more detail must be added to this methods subsection, it is unclear what type of experiment the authors did. In looking up the cited papers it appears that they did mass spectrometry – however this needs to be described in the current manuscript as it was not even mentioned.

This section was expanded adding a summary of the about the proteomic analysis, since this is described in Porras-Agüera et al. (2019). Lines 97-104.

-It is insufficient to perform mass spectrometry without examining select protein levels by western blot. The authors need to choose a subset of proteins to validate the expression of by western blot analysis. It should not be difficult to obtain cerevisiae antibodies to some of these proteins.

This untargeted approach aims to determine the whole cell wall proteome under CO2 overpressure. Those proteins showing highest differences with the control condition will be subjected to further experiments such as validation, genetic and electron microscopic studies and other have been planned for future publications.

-From the way the manuscript is written, it appears as though the actual experiments were published in (Porras-Aguera et al., 2019 Microorganisms; https://doi.org/10.3390/microorganisms7110542) and that this paper is simply an additional analysis of this dataset with absolutely no new experiments. This does not constitute enough for a separate manuscript and the authors must greatly expand this paper in order for it to be suitable as a standalone manuscript. Perhaps the authors could choose some key genes of interest, knock them out and examine the effects on wine production. There are countless ways this manuscript can be expanded and I will not list them all, however this manuscript must be expanded somehow. The authors can use the mass spec results to inform their further analyses.

The experimental design is described in Porras-Agüera et al. (2019), but the objective of the manuscript and the gene ontology is different. This manuscript represents a part of a bigger project focused on knowing the cell response at proteomic level of wine yeast strains to CO2 overpressure, typical conditions of sparkling wines elaboration. The current study is a first and novel proteomic approach in order to characterize the cell wall response to pressure in two industrial wine strains. Authors believe that the manuscript provides with potential cell wall biomarkers that need to be verified with other studies and using other approaches. However, these studies have been planned for future publications.

Minor points

-CO2 in title should either be formatted with a subscript “2” or spelled out as carbon dioxide

The title was modified changing the word CO2.

-Line 27: “Flocculation-related protein was not relevant under pressure conditions”, what is meant by not relevant?

This means that proteins associated with flocculation were not identified under pressure conditions. They were obtained with low levels under non-pressure conditions. This has been corrected in the manuscript.

-Line 67: “a first proteomic approach” should be “a novel proteomic approach”

The sentence was changed.

-Line 88: “o” should be “or”

The word was modified.

-Lines 96-100: paragraphs must be at least 3 sentences in length

The paragraph was expanded adding a summary of the proteome analysis. Lines 97-104.

-Line 104: “down-represented” should be “under-represented”

The expression was modified along the text. Lines 108 and 171.

-Line 106: the “GO” acronym must be defined (“Gene ontology”)

The acronym was defined.

-Reference to the “p” in “p-values” and “p < 0.XXX” must be italicized

The letter was italicized along the manuscript.

Round 2

Reviewer 3 Report

Thank you for the edits you have made. They have improved the manuscript. I understand the experiments I suggested are to be included in future studies and will defer to the editor on whether or not they believe the experiments are required for the current manuscript or not.